

# Brief communication: Impacts of ocean-wave-induced breakup of Antarctic sea ice via thermodynamics in a standalone version of the CICE sea-ice model

Luke G. Bennetts[1], Siobhan O'Farrell[2], and Petteri Uotila[3]

[1]School of Mathematical Sciences, University of Adelaide, Adelaide, SA, Australia
[2]CSIRO Ocean and Atmosphere, Aspendale, VIC, Australia
[3]Finnish Meteorological Institute, Helsinki, Finland

*Correspondence to:* L. G. Bennetts (luke.bennetts@adelaide.edu.au)

**Abstract.** Impacts of wave-induced breakup of Antarctic sea ice on ice concentration and volume are investigated using a modified version of the CICE sea ice model, run in standalone mode from 1979–2010. Model outputs show that breakup reduces ice concentration by up to 0.3–0.4 in a vicinity of the ice edge during the summer, and total ice volume by over $500 \, \mathrm{km}^3$.

## 1 Introduction

Speculation surrounding the impacts of ocean surface waves on the world's sea ice is building. In the Antarctic, the speculation has been fuelled by Kohout et al. (2014)'s findings that trends in ice-edge contraction (from satellite observations) are closely correlated to trends in increasing local significant wave heights (from a numerical model) and, conversely, trends in ice-edge expansion are correlated to trends in decreasing significant wave heights. They attributed these correlations to large-amplitude storm waves propagating into the ice-covered ocean and breaking up the ice cover into relatively small floes, which are more mobile and vulnerable to melting. This relationship can be inferred from descriptions of the way in which waves regulate the morphology of the ice cover in the first 10s to 100s of kilometres in from the ice edge, originally made by Squire, Wadhams and co-workers in the 1970s (see, e.g., the review Squire et al., 1995) — a region often referred to as the marginal ice zone, although the term is not adopted in this study due to ambiguity in its definition. Kohout et al. (2014) suggested that incorporating wave impacts on sea-ice into climate models will empower the models to capture sea ice responses to climate change, for example, the regional variability of trends in Antarctic sea-ice extent (Stammerjohn et al., 2008).

This study constitutes the first quantification of Antarctic sea-ice breakup by waves on ice concentration and volume. It uses a standalone version of the CICE sea-ice model, modified to include wave-induced breakup, with wave forcing provided by a Wavewatch III wave-model hindcast in ice-free grid cells close to the ice edge. Wave energy advects into cells containing ice cover, where models of wave-energy attenuation due to ice cover and wave-induced ice breakup are applied, in a similar manner to the operational ice/ocean model wave–ice interaction component developed by Williams et al. (2013a, b).





CICE v4.1 (Hunke and Lipscomb, 2010) is used for the study, in which floe diameters appear in the lateral ice-melt model only, and are set to be 300 m throughout the ice cover by default. Breakup reduces mean floe diameters typically to 20–100 m in cells extending ∼100 km in from the ice edge, beyond which the wave energy is no longer strong enough to break the ice. When ocean temperatures are high enough to melt ice, the reduced diameters promote lateral melt, reducing the ice concentration,

which, in turn, reduces the ice strength, so that breakup indirectly impacts both ice concentration and volume through dynamic processes. Model outputs show that during the summer wave-induced breakup reduces ice concentration by up to 0.3–0.4 and total ice volume by $> 500\,\mathrm{km}^3$. During the winter, the ice concentration recovers, but volume changes persist, becoming dispersed over the inner ice pack.

## 2  Model

CICE uses an ice-thickness-distribution function $g(\mathbf{x}_{ij}, t : h)$ to describe the sea-ice cover, in which $\mathbf{x}_{ij}$ denotes a grid cell on the ocean surface, indexed $i$ in longitude and $j$ in latitude, $t$ denotes time, and $h$ denotes ice thickness, with $g(\mathbf{x}_{ij}, t : h)dh$ the fractional area of ice in cell-$ij$ with thickness in the interval $(h, h + dh)$. The ice-thickness distribution is calculated as a numerical approximation of the ice-thickness-evolution equation

$$\frac{\partial g}{\partial t} = -\nabla \cdot (g\mathbf{u}) - \frac{\partial}{\partial h}(fg) + \psi, \tag{1}$$

using discrete time steps with a nominal global step length of one hour, a horizontal tripolar grid with a nominal resolution of one latitudinal/longitudinal degree, and partitioning of the ice into discrete thickness categories (five categories plus open water are used for this study, as standard). The first term on the right-hand side of Eqn. (1) denotes ice advection, where $\mathbf{u}$ is ice velocity, calculated via the elastic–viscous–plastic (EVP) rheology model of Hunke and Dukowicz (1997). The second term denotes thermodynamic thickness redistribution, where $f$ is the rate of melting or freezing. The final term denotes mechanical

redistribution due to ridging.

Waves are introduced into the model using the wave-energy-density spectrum, $S(\mathbf{x}_{ij}, t : \omega, \theta)$, where $\omega$ and $\theta$ denote angular frequency and wave direction, respectively. This is the standard description of waves in oceanic general-circulation models. At the beginning of each time step, incident spectra are prescribed in grid cells at a latitude outside the ice cover but as close to the ice cover as possible. For expediency, in each cell at the incident latitude, the wave field is set to be a Bretschneider

spectrum, defined by a significant wave height and a peak period, propagating in the mean wave direction. In subsequent cells, directions are calculated as averages of the wave directions entering the respective cells, weighted according to the associated wave energy.

Assuming steady-state conditions over a time step, the spatial distribution of wave energy in the ice-covered ocean is calculated according to a discrete version of the wave-energy-balance equation

$(\cos\theta, \sin\theta) \cdot \nabla S = -\alpha S. \tag{2}$

The attenuation coefficient, $\alpha(\mathbf{x}_{ij}, t : \omega)$, is set as

$\alpha = \alpha_0 \equiv c(\beta_2 \omega^2 + \beta_4 \omega^4) \quad \text{where} \quad \beta_2 \approx 7.68 \times 10^{-5} \quad \text{and} \quad \beta_4 = 4.21 \times 10^{-5}, \tag{3}$



based on Meylan et al. (2014)'s empirical model, scaled according to the areal concentration of sea ice on the ocean surface, $c(\mathbf{x}_{ij}, t)$.

In each cell, the floe-size distribution is defined by a representative floe diameter $D(\mathbf{x}_{ij})$, for consistency with the assumptions underlying the lateral-melt model, described below. At the beginning of a simulation, the diameters are set to the relatively large value $D(\mathbf{x}_{ij}) = D_{\mathrm{mx}} = 300 \, \mathrm{m}$, for consistency with the value used throughout the ice cover in existing versions of CICE. For cells in which wave energy is non-negligible, Williams et al. (2013a)'s ice-breakup criterion is applied, with the diameter of the broken floes denoted $D_{\mathrm{bk}} < D_{\mathrm{mx}}$. Following Bennetts et al. (2015), the representative floe diameter in cell-$ij$ post wave-induced breakup is calculated as a weighted average of the broken-floe diameter over the fraction of the cell where the waves are strong enough to cause breakup, $a_{\mathrm{bk}}$, and the diameter in the cell at the beginning of the time step, $D_0$, in the remaining fraction, i.e. $D(\mathbf{x}_{ij}) = a_{\mathrm{bk}}(\mathbf{x}_{ij})D_{\mathrm{bk}}(\mathbf{x}_{ij}) + (1 - a_{\mathrm{bk}}(\mathbf{x}_{ij}))D_0(\mathbf{x}_{ij})$. For cells at the outermost fringes of the ice-covered ocean, where the ice is too thin and compliant to be broken by waves, the floes diameters are assumed to be small, and assigned the representative diameter $D = D_{\mathrm{mn}}$.

In cells where breakup occurs, the broken-floe diameter $D_{\mathrm{bk}}$ is calculated by assuming the in-cell floe-size distribution obeys a split power law, as observed by Toyota et al. (2011), and with Williams et al. (2012)'s probability-density function $p(d)$, where $d$ denotes floe diameter, defined by

$$p(d) = \frac{\mathbb{P}_0 \beta_0 \gamma_0}{d^{\gamma_0 + 1}} \qquad \text{if} \quad d \in [D_{\mathrm{mn}}, D_{\mathrm{cr}}), \qquad \text{where} \quad \beta_0 = \left( D_{\mathrm{mn}}^{-\gamma_0} - D_{\mathrm{cr}}^{-\gamma_0} \right)^{-1}, \tag{4a}$$

$$p(d) = \frac{(1 - \mathbb{P}_0) \beta_1 \gamma_1}{d^{\gamma_1 + 1}} \qquad \text{if} \quad d \in [D_{\mathrm{cr}}, \infty), \qquad \text{where} \quad \beta_1 = D_{\mathrm{cr}}^{\gamma_1}, \tag{4b}$$

and $p(d) = 0$ if $d < D_{\mathrm{mn}}$. Here, $D_{\mathrm{mn}}$ represents a minimum floe diameter, which is chosen to be equal the diameter for small floes for simplicity; $D_{\mathrm{cr}}$ is a critical diameter marking the transition from small to large floes (found to be in the range 15–40 m by Toyota et al., 2011), and $\gamma_1 = 1.15$ and $\gamma_2 = 2.5$ are representative exponents for small- and large-floe regimes, respectively (Toyota et al., 2011). The quantity $\mathbb{P}_0 \in [0, 1]$ weights the distribution towards small floes (large $\mathbb{P}_0$) or large floes (small $\mathbb{P}_0$). Its value is set as

$$\mathbb{P}_0 = 1 - q \left( \frac{D_{\mathrm{pr}}}{D_{\mathrm{cr}}} \right)^{\gamma_1} \qquad \text{where} \qquad D_{\mathrm{pr}} = \lambda / 2 \qquad \text{is the predicted breakup diameter,} \tag{5}$$

equal to the distance between successive strain maxima for a regular wave train at the dominant wavelength $\lambda$ for the spectrum $S$, propagating through a uniform floe (Williams et al., 2013a; Bennetts et al., 2015), so that a chosen proportion $q$ of floe diameters are greater than $D_{\mathrm{pr}}$. In the uncommon event that $D_{\mathrm{pr}} < D_{\mathrm{cr}}$ then $\mathbb{P}_0 = 0$, noting that $D_{\mathrm{cr}}$ approximates the theoretical diameter below which flexural breakup cannot occur (Toyota et al., 2011). The broken-floe diameter $D_{\mathrm{bk}}$ is the mean diameter in a given cell, i.e.

$$D_{\mathrm{bk}} = \int_{D_{\mathrm{mn}}}^{\infty} pD \, \mathrm{d}D = \frac{\mathbb{P}_0 \gamma_0 \beta_0 (D_{\mathrm{mn}}^{1-\gamma_0} - D_{\mathrm{cr}}^{1-\gamma_0})}{\gamma_0 - 1} + \frac{(1 - \mathbb{P}_0) \gamma_1 \beta_1 D_{\mathrm{cr}}^{1-\gamma_1}}{\gamma_1 - 1}. \tag{6}$$

The breakup model is applied at the beginning of each CICE time-step, allowing the reduced floe diameters to affect other CICE-model components. The reduced diameters directly affect the contribution of lateral melting, $r_{\mathrm{lat}}$, to reducing the ice





concentration via the discrete version of Steele (1992)'s model

$$r_{lat} = \frac{\pi \Delta t w_{lat}}{\mu D}, \tag{7}$$

which assumes floes in a given cell are identical. Here $\mu = 0.66$ is a geometric parameter, and $w_{lat} = 1.6 \Delta T^2 \times 10^{-6}$ is the rate of lateral melt, in which $\Delta T$ is the temperature difference of the sea surface above that of the bottom of the ice (set to

zero if the difference is negative). The diameter is updated at the end of the thermodynamic routine to account for lateral melt. The floe-diameter parameter is a tracer field in CICE, and is transported within each ice category to give the total floe-size distribution at the end of a time step.

During the summer months, when the ice is weaker and towards its minimum extent, waves cause breakup close to the coastline. The existing thermodynamic models in CICE do not increase the diameters of these broken floes fast enough through

the winter to create a realistic seasonal cycle for the floe-diameter distribution. Therefore, an ad-hoc floe-bonding scheme is applied, in which the floe diameter in a given cell is doubled if the freezing potential in that cell is positive, up to the maximum diameter $D_{mx}$.

## 3   Results

The model was run from 1979–2010 using input wave data generated by a Wavewatch III model hindcast (Durrant et al.,

2013), and atmospheric and oceanic data from the U. S. National Center for Environmental Prediction's Climate Forecast System Reanalysis (NCEP's CFSR, Saha et al., 2010). The minimum and critical floe diameters are set as $D_{mn} = 5\,\text{m}$ and $D_{cr} = 30\,\text{m}$, and, following breakup, the proportion $q = 0.05$ of floe diameters are set to be greater than the predicted breakup diameter $D_{pr}$.

Fig. 1 shows example model outputs for two dates during 1995 (i.e. a year half-way through the simulation), representative

of results in summer (1st January, panels in top row) and winter (1st July, bottom). The panels in the left-hand column show significant wave heights, with the sharp outer boundaries indicating the latitudes at which data is extracted from the wave model. This boundary is farther north in the winter because the ice extent is greater than in the summer. The regions of rapid wave-height decrease with respect to southward distance indicate attenuation of wave energy due to ice cover. In the summer, packets of wave energy are able to propagate almost to the coastline, particularly around the Antarctic peninsula, due to reduced

ice cover in that locality.

The middle column shows the extent of ice coverage, with the ice divided into regions according to floe size. Regions of small diameter floes (green) are identified as those cells for which $D \leq D_{mn} = 5\,\text{m}$, wave-broken floes (red) are the floe-size interval $D_{mn} < D \leq 250\,\text{m}$, and unbroken floes (grey) are $D > 250\,\text{m}$. The right-hand column shows the impact of the small and broken floes on ice concentration, in terms of the difference in concentration between the simulation without breakup

($D = 300\,\text{m}$) and the simulation with breakup, with positive values indicating decreases in concentration due to breakup.

The Southern Ocean experiences the strongest waves during winter, as indicated in the left-hand column. However, the regions of broken ice are comparable between the two seasons (approximately 10 % smaller in the summer), as the lower





**Figure 1.** Example model outputs using minimum and critical floe diameters $D_{mn} = 5\,\text{m}$ and $D_{cr} = 30\,\text{m}$, and $q = 0.05$. The left-hand column shows the significant wave heights. The middle column shows the ice regions: small floes (green), wave-broken floes (red), unbroken floes (grey) and no ice/open water (blue). The right-hand column shows the change in concentration between the simulations without and with breakup. The top row is representative of results in austral summer and the bottom row of winter.

summer ice concentration allows waves to penetrate deeper into the ice-covered ocean, relative to their incident energy. The ice is structured into approximately uniform bands in the winter, whereas in the summer coastal effects complicate the structure.

In the summer, the broken ice decreases the ice concentration in a vicinity of the ice edge, with reductions of $\sim 0.1$ common, but numerous pockets of 0.3–0.4 reductions apparent. The region most impacted by breakup is estimated by the region bounded by the two black lines, where the outer black line denotes the first cell (with respect to each longitude) at which the ice concentration exceeds 0.1, and the inner black line represents three cells further in (or land if that begins before the third cell).




During the winter, the concentration change is too small to be visible on the scale shown (order 0.01), as the temperatures are too low to melt the broken floes.

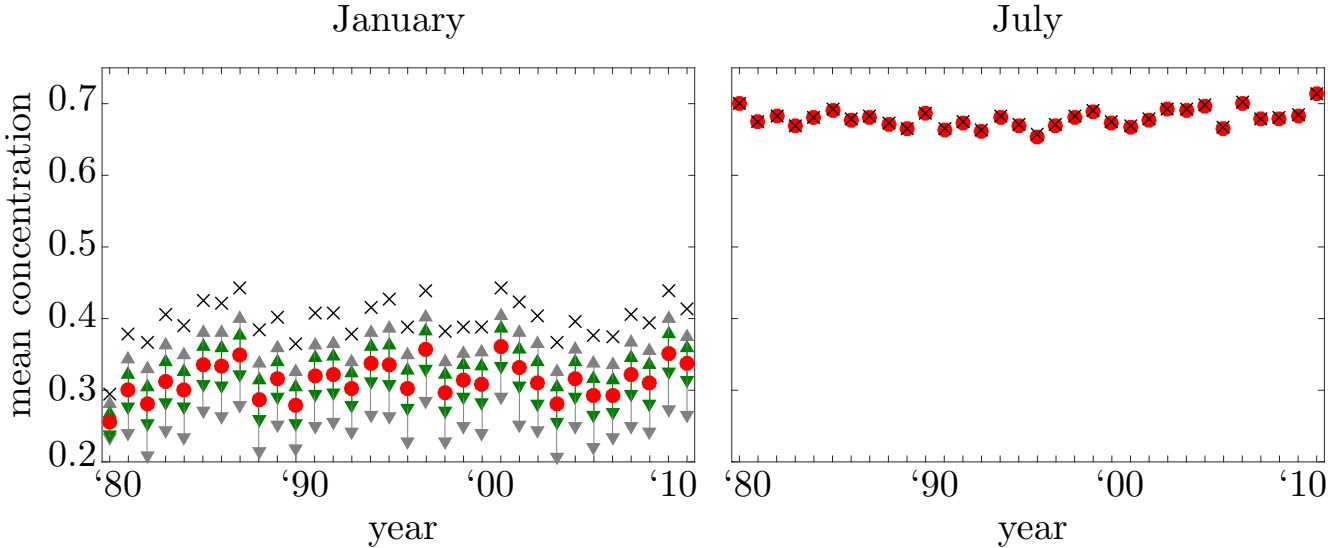

**Figure 2.** The mean–monthly ice concentration at the ice edge, for January (left-hand panel) and July (right). Results are for the simulation without breakup ($\times$) and with breakup for: the parameters considered in Fig. 1 ($D_{\mathrm{mn}} = 5\,\mathrm{m}$, $D_{\mathrm{cr}} = 30\,\mathrm{m}$, and $q = 0.05$, •); smaller floes $D_{\mathrm{mn}} = 2.5\,\mathrm{m}$, $D_{\mathrm{cr}} = 20\,\mathrm{m}$ and $q = 0.025$ (▼); larger floes $D_{\mathrm{mn}} = 10\,\mathrm{m}$, $D_{\mathrm{cr}} = 40\,\mathrm{m}$ and $q = 0.1$ (▲); an increased attenuation rate $\alpha = 10\alpha_0$ (▼); and a decreased attenuation rate $\alpha = \alpha_0/10$ (▲).

Fig. 2 shows mean–monthly ice concentrations at the ice edge (the region bounded by the black lines in the right-hand column of Fig. 1), for each simulation year. Results are again shown for January and July, as representations of summer and winter conditions, respectively. Data were generated for simulations without breakup ($\times$) and with breakup (•). For the summer conditions, additional data indicate sensitivities of concentration changes to: (i) the floe-size parameters, with data given for simulations in which $D_{\mathrm{mn}}$, $D_{\mathrm{cr}}$ and $q$ are decreased to $D_{\mathrm{mn}} = 2.5\,\mathrm{m}$, $D_{\mathrm{cr}} = 20\,\mathrm{m}$, and $q = 0.025$ (▼) and increased to $D_{\mathrm{mn}} = 10\,\mathrm{m}$, $D_{\mathrm{cr}} = 40\,\mathrm{m}$ and $q = 0.1$ (▲); and (ii) increasing or decreasing the wave-attenuation coefficient, $\alpha$, by an order of magnitude ($\alpha = 10\alpha_0$, ▼, and $\alpha = \alpha_0/10$, ▲, respectively). The ranges of floe sizes and attenuation rates are within the limits of present uncertainty.

As indicated by Fig. 2 and the bottom–left panel of Fig. 1, breakup has negligible impact on ice concentration during winter. During the summer, breakup reduces the concentration, with the mean decrease being $\sim 0.08$ for the parameters used in Fig. 1 (neglecting the first, spin-up year of the simulation). Reducing the floe-size parameters increases the impact of breakup (as smaller floes melt more rapidly than larger ones), and increasing them reduces the impact, with the mean reductions compared to the simulation without breakup being $\sim 0.11$ and $0.06$, respectively. Similarly, reducing the attenuation rate increases the





impact (as the waves maintain their strength for greater distances into the ice-covered ocean), and increasing the attenuation rate has the opposite effect — the mean reductions are $\sim 0.15$ and $0.04$, respectively.

The top panels of Fig. 3 show changes in ice volume due to breakup, for the two dates used in Fig. 1, i.e. results representative of summer (1st January 1995, left-hand panel) and winter (1st July 1995, right). During the summer, breakup decreases the ice volume, with losses of up to $\sim 2.7\,\mathrm{km}^3$ in individual cells. The pattern of the decreases is strongly correlated with the concentration decreases shown in the top–right panel of Fig. 1. However, reductions in ice thickness forced by dynamic processes also contribute to volume losses with mean in-cell thicknesses up to $0.96\,\mathrm{m}$ thinner with breakup for the date shown.

During the winter, volume losses of $\sim 0.5\,\mathrm{km}^3$ per grid cell (but up to $1\,\mathrm{km}^3$) are visible in the interior of the ice cover (the unbroken ice region). This contrasts with the negligible concentration losses on the same date shown in bottom–left panel of Fig. 1. The volume losses result from summer thickness reductions being restored at a slower rate than concentration. Ice advection disperses the losses over large regions.

The bottom–left panel of Fig. 3 shows mean–monthly decreases in ice volumes due to breakup, over a typical six-year interval. The ice volumes are sums over the total ice cover (for cells with concentrations greater than 0.1, ◆) and cells at the ice edge (the region between the black lines, ∗). Seasonal cycles are evident, with, for example, maximum total volume losses of $600$–$760\,\mathrm{km}^3$ occurring in December and minimum losses of $260$–$320\,\mathrm{km}^3$ occurring in August. Losses at the ice edge are negligible during winter, but are up to $470\,\mathrm{km}^3$ during summer, accounting for increases in total volume loss during that season.

The bottom–right panels of Fig. 3 show decreases in total ice volume per latitude on 1st January (bottom panel) and 1st July (top), over the full 32 years of the simulations, in terms of the median values, and the spread, in terms of the 25th and 75th percentiles. Data are split into losses in the eastern (—) and western (—) sectors of Antarctica. During the summer, the losses in the two sectors are similar. During the winter, western-sector losses outweigh those of the eastern sector, with median losses for the western sector on average $0.57\,\mathrm{km}^3$ greater than the eastern sector. This is attributed to a significant proportion of the East Antarctic sea ice that is impacted by breakup during the summer, melting during February, so that the winter ice is largely composed of new ice, with no memory of the breakup.

## 4  Discussion

The findings of this pilot study indicate that increased lateral ice melt over the first $\sim 100\,\mathrm{km}$ in from the ice edge, due to small wave-broken floes, and the follow-on effects on ice dynamics, impact ice concentration and volume in a vicinity of the edge during winter, and ice volume in the interior pack throughout the year. Horvat et al. (2016)'s coupled ice–ocean–atmosphere model, which includes interactions between floe diameters, ocean circulation and ice melt, indicates that lateral melt remains important for sea-ice evolution for floe diameters orders of magnitude larger than the $O(30\,\mathrm{m})$ limit given by Steele (1992)'s model, as used in CICE. Presumably, therefore, integrating diameter–circulation–melt interactions into the modified version of CICE would strengthen the impacts of breakup. Moreover, integrating Feltham (2005)'s granular floe-size dependent rheology would provide a direct impact of breakup on ice dynamics. Applying the modified CICE model in a fully-coupled setting will unlock feedbacks triggered by the breakup — for example, the reduced concentration due to increased lateral melt releasing





**Figure 3.** Top row: Snapshots of ice volume changes between simulations without and with breakup ($D_0 = 5\,\mathrm{m}$, $D_{\mathrm{cr}} = 30\,\mathrm{m}$ and $q = 0.05$). Bottom–left panel: Mean–monthly decreases in ice volume, over total ice cover (◆) and at ice edge (∗), for 1990–1995. Bottom–right panels: Median decrease in ice volume per latitude for the eastern sector (—) and the western sector (—), on 1st January (bottom panel) and 1st July (top) for all simulation years. Shaded regions show corresponding 25th to 75th percentile ranges.



more oceanic heat to the atmosphere, thus increasing upwelling of ocean heat through convection and hence promoting further ice melt — permitting studies into influences on long-term trends in ice concentration, volume and also extent. If the community judges the impacts of floe-size dependent processes to be significant, future large-scale sea-ice models may be developed along the lines of the theories for coupled ice-thickness and floe-size evolution outlined by Zhang et al. (2015) and Horvat and

5  Tziperman (2015).

## 5  Code availability

The Australian Antarctic Data Centre hosts the code used for this study at doi:10.4225/15/57D0EA42ED985.

*Acknowledgements.*  The authors thank Mark Hemer for providing advice on the wave data used. The Australian Research Council (DE130101571) and the Australian Antarctic Science Program (Project 4123) funded this investigation. The Academy of Finland supports PU (Contract

10  264358).



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
