# Peer review of "Brief communication: Impacts of ocean-wave-induced breakup of Antarctic sea ice via thermodynamics in a standalone version of the CICE sea-ice model"

_The Cryosphere, 2016_

## Referee Comment (RC1) · Anonymous Referee #1 · 16 Jan 2017

General comments This paper is clearly written and figures are a clear representation of the results. The paper is a useful pilot study highlighting the potential benefits of inclusion of waves in a sea ice model. More work would be required to make any stronger statements.

Specific comments 1. Does the paper address relevant scientific questions within the scope of TC? Yes, the topic is currently relevant and will interest a significant number of research groups world wide. 2. Does the paper present novel concepts, ideas, tools, or data? Yes, actively including waves within CICE with a focus on Antarctica is novel.

3. Are substantial conclusions reached? This work further highlights the potential for waves to be an important component in CICE in the southern hemisphere. It however does not show that it is. The modelling work would need to be fully coupled and compared against observations to show an improvement over CICE. 4. Are the scientific methods and assumptions valid and clearly outlined? The methods and assumptions are well articulated and appear to be valid. 5. Are the results sufficient to support the interpretations and conclusions? This paper does not overstate the results and highlights that this paper is a pilot study to motivate further research. The results sufficiently show that the modified model, given the assumptions and initial conditions, has the capacity to have an impact on sea ice in summer. 6. Is the description of experiments and calculations sufficiently complete and precise to allow their reproduction by fellow scientists (traceability of results)? The code used in this study is available and the paper is described in such a way that the study should be able to be reproduced. 7. Do the authors give proper credit to related work and clearly indicate their own new/original contribution? The authors give appropriate credit to related work and articulate the new contribution they are making. 8. Does the title clearly reflect the contents of the paper? The title is appropriate. 9. Does the abstract provide a concise and complete summary? yes 10. Is the overall presentation well structured and clear? yes 11. Is the language fluent and precise? yes 12. Are mathematical formulae, symbols, abbreviations, and units correctly defined and used? yes 13. Should any parts of the paper (text, formulae, figures, tables) be clarified, reduced, combined, or eliminated? no 14. Are the number and quality of references appropriate? yes 15. Is the amount and quality of supplementary material appropriate? yes

Technical corrections

None spotted

---

## Short Comment (SC1) · 31 Jan 2017

Dear authors,

The work shown here is extremely exciting. The inclusion of wave-breaking and a true floe thermodynamics into CICE is an important step towards improving sea ice models, and I look forward to future work implementing this model. I generally find the communication to be useful, but I wanted to bring up an important, and subtle, issue that I feel should be addressed in this communication and going forward. On pg. 4 line 5,

"The floe-diameter parameter is a tracer field in CICE, and is transported within each ice category to give the total floe-size distribution at the end of a time step"

The mean floe diameter, however, does not advect as a tracer. In the parameterization of lateral melting based on *Steele* (1992), as well as the framework presented here, the mean floe diameter is computed as the average floe diameter across all of the floes within a grid cell, i.e.

$$\overline{D} = \frac{1}{N} \sum_{1}^{N} d_i = \int_{D_{min}}^{\infty} p(D)DdD, \tag{1}$$

where $\{d\}_i$ is the collection of $N$ floe diameters, and I use the probability density function notation from this manuscript, where $\int p(D)dD = 1$. In general, the quantity $p(D)dD$ must be equal to the number of floes per unit area with diameter between $D$ and $D + dD$, which I call $N(D)dD$, divided by the number of floes per unit area, $\mathcal{N}$, i.e.

$$p(D) = \frac{N(D)}{\mathcal{N}}. \tag{2}$$

By definition, $\mathcal{N} = \int N(D)dD$ and so $p$ is properly normalized. Both $N(D)$ and $\mathcal{N}$ advect as tracers with the two-dimensional ice-velocity field. The probability density function, however, as the ratio of these two terms, will not. For this reason, the mean floe diameter also does not advect as a tracer. This can lead to pathologies in sea ice evolution (*Horvat and Tziperman*, 2017). The attached figure shows the evolution of normalized four state variables which are advected from an adjacent grid cell into and through a single grid cell. The two adjacent cells have different floe number, concentration, and mean thickness at t=0. In this case the full FSTD (floe size and thickness distribution) is computed and solved for at each model timestep. In the plot, 0 corresponds to the initial value, and 1 corresponds to the value from the adjacent grid cell.

Mean floe size has (and incidentally, mean ice thickness) has a different evolution than does concentration and volume, which have an exponential approach.

The simple explanation for this is that $p$ has a normalization by a time-varying quantity (the total floe number). To properly include mean floe size within CICE, one has to account for the evolution of $\mathcal{N}$ in addition to $p$. This can be done by observing,

$$c = \mathcal{N} \int_{D_{min}}^{\infty} \frac{\pi}{4} D^2 p(D) dD, \tag{3}$$

where $c$ is the ice concentration. Both $p$ and $c$ are available from the model, so $N(D)$ and $\mathcal{N}$ can be computed, and hopefully no major modifications to the model code are necessary. Quite possibly the proper mean floe size advection scheme is unimportant, but as you are the first to introduce this type of model, it is unclear, and is exciting to find out. If future models include a fully-evolving FSD, this fix will no longer be necessary.

I can provide more information, or a more mathematical derivation of the aforementioned pathologies if this point is not well-made here!

**References**

Horvat, C., and E. Tziperman (2017), The evolution and emergence of scaling laws in the sea ice floe size distribution, *J Geophys Res*, *In Review*.

Steele, M. (1992), Sea ice melting and floe geometry in a simple ice-ocean model, *J. Geophys. Res. Oceans*, *97*(C), 17,729, doi:10.1029/92JC01755.

[Figure]

[Figure: plot of "Approach" (y-axis, 0 to 1) vs "Time (days)" (x-axis, 0 to 12) showing four curves with legend: Concentration (red solid), Volume (black dashed), Mean Thickness (black solid), Mean Floe Size (red dashed)]

**Fig. 1.** Approach of several advected model variables

---

## Referee Comment (RC2) · Anonymous Referee #2 · 16 Feb 2017

This paper starts with the CICE sea-ice model and adds waves that break up the sea ice in grid cells near the ice edge, causing that ice to melt more quickly in summer due to greater lateral surface area.

In general the paper is well conceived and the mechanism is plausible. My comments (below) are minor. Comments are given in page order, not in order of importance. I have not read the other reviews of this paper that are posted on TCD web site, in order to maintain an independent review here.

Comments

[Figure]

Page 1, Abstract. Total ice volume is reduced by over 500 km^3, but this needs to be put into context. What is the volume of the entire ice cover?

Page 2, lines 6-7. Same comment as above.

Page 2, equation (3). What are the units of beta_2 and beta_4? From equation (2), alpha must have units of 1/length. The sea-ice concentration, c, is dimensionless, and omega has units of 1/time, so beta_2 must have units of time^2/length, and beta_4 must have units of time^4/length. This should be explicitly noted.

Page 3, line 10. How is a_bk(x) chosen or determined? There is nothing about it in the rest of the paper.

Page 3, lines 13-14. The assumption here is that the floe size distribution (FSD) follows a split power law, with one exponent for floes smaller than a critical size and another exponent for floes larger than the critical size, following Toyota et al (2011). This is a dubious formulation of the FSD. First of all, if one actually looks at figure 9 of Toyota et al (2011), one sees that the FSD is a continuously curving concave-down shape, rather than two power-law regimes. This was noted in earlier work by Herman (2010), who wrote in reference to an earlier paper by Toyota: "However, contrary to how the above authors interpret their results, in both cases the change in slope of the FSD seems rather gradual than abrupt. Instead of a combination of two power laws "glued together" at a highly arbitrarily chosen floe diameter, another type of distribution would be desirable. It should reflect the observed gradually increasing deviation from a power-law distribution for decreasing floe diameter." Herman, A. (2010), Sea-ice floe-size distribution in the context of spontaneous scaling emergence in stochastic systems, Physical Review E, 81, DOI: 10.1103/ PhysRevE.81.066123 Furthermore, other researchers have found power-law behavior for the Antarctic FSD in which the exponent changes as the ice edge is approached, but without a critical floe size separating two power-law regimes (Paget et al, 2001; Lu et al, 2008). Returning to the current paper, a much simpler assumption for the FSD would have been a simple power law

with one exponent. This would have eliminated the need for four parameters: $D_{cr}$, $q$, gamma_1, and P0. It would be interesting to know whether the results hold up under this simpler (and possibly more realistic) FSD. However, I would not insist that the authors re-do all their calculations, unless it's a simple thing to do (maybe just set $D_{cr} = D_{mn}$ and P0 = 0). But they should acknowledge that their results rest on the questionable split power-law formulation of the FSD.

Page 3, equation (5). In the definition of P0, there is an exponent gamma_1. Is this the same gamma_1 as in equation (4b)? Why should it be the same exponent as in the probability density function? I don't understand the reasoning or the math for the use of gamma_1 here.

Page 4, equation (7) and following. What are the units of r_lat and w_lat? What is the value of the time step delta_t?

Page 6, lines 8-9, and Figure 2 (left panel). The text and the figure indicate that LESS attenuation of waves results in HIGHER mean ice concentration. I would have thought that less attenuation would allow more wave energy to penetrate into the ice pack and break up the ice, resulting in lower ice concentration. Please explain why less attenuation leads to higher ice concentration, and more attenuation leads to lower ice concentration.

Page 6, line 15. "reducing the attenuation rate increases the impact..." I agree that reducing the attenuation rate SHOULD increase the impact of the waves, but Figure 2 shows that reducing the attenuation rate actually reduces the impact of the waves. The symbols for reduced attenuation rate (upward-pointing triangles) are much closer to the no-break-up case (crosses) than the symbols for increased attenuation rate (downward-pointing triangles). This doesn't make sense to me.

Page 7, line 8. "volume losses of $\sim$0.5 km^3 per grid cell" – This doesn't mean anything unless we know how big a grid cell is. On page 2, lines 15-16, we are told that the nominal resolution of the grid is 1 degree in latitude and 1 degree in longitude. The

extent of 1 degree of longitude depends on latitude, so the size of a grid cell (in km^2) depends on latitude. At the latitude of the Antarctic Circle, 1 degree of longitude is about 44 km. So I calculate that the area of a grid cell is roughly 111 x 44 = 4884 km^2. A volume of 0.5 km^3 of ice spread over such a grid cell is about 10 cm of ice thickness (at 100% concentration) or 20 cm of ice thickness (at 50% concentration). So now I can understand roughly what a volume loss of 0.5 km^3 per grid cell means. Please help out the reader by providing this kind of information.

Page 7, lines 20-23. If I understand this correctly, the eastern sector contains mostly first-year ice ("new ice"), with no memory of break-up, while the western sector presumably contains some multiyear ice, which retains memory of break-up?

Technical Comments

Page 1, Abstract, line 2. "Model output shows that WAVE-INDUCED breakup..."

Page 2, equation (1). Cite Thorndike et al (1975), The Thickness Distribution of Sea Ice, JGR.

Page 3, line 18. "which is chosen to be equal TO the diameter..."

Page 3, line 20. Values are given for gamma_1 and gamma_2, but they should probably be gamma_0 and gamma_1. There is no gamma_2 in the equations.

Page 3, line 25. propagating through a uniform floe FIELD (?)

Page 3, equation (6). Inside the integral, "pD" should be "p(D)" i.e. put parentheses around the "D"

Page 4, lines 21-22. In reference to Figure 1, in the left-hand panels showing wave height, it looks to me like the "sharp outer boundaries indicating the latitudes at which data is extracted from the wave model" are at the same latitude in each panel. But the next sentence says, "The boundary is farther north in winter..." I don't see that the outer boundary of extracted data is farther north in the bottom panel. The outer circular

boundary appears to be at exactly the same latitude in both panels. If the authors are referring to an INNER boundary that is several grid cells inside the outer boundary, they should mark it more clearly.

Page 4, line 32. This sentence should refer to the middle column of Figure 1, just as the previous sentence refers to the left-hand column. Otherwise the reader may not shift her/his attention to the middle column.

Page 5, Figure 1, upper right panel showing concentration change. This panel is a bit too small – it's hard to see the regions of large change. A figure the size of Fig 3 would be better.

Page 7, line 9. "bottom-left panel" should be "bottom-right panel"

Page 7, line 17. "ice volume per latitude" should be "ice volume per degree of latitude". Similarly in Figure 3 (in the title of the bottom right panel) and in the caption.

Page 8, Figure 3. The eastern and western sectors should be marked in the upper panels (the maps).

Page 9, lines 2-3. "If the community judges the impacts..." – maybe better to say "If further research finds the impacts..."

Page 11. "Schwinger" should be "Schweiger"

---

## Author Comment (AC2) · 17 Mar 2017

Dear Chris

*The work shown here is extremely exciting. The inclusion of wave-breaking and a true floe thermodynamics into CICE is an important step towards improving sea ice models, and I look forward to future work implementing this model.*

Thanks for your interest in our work and your useful comment.

*I wanted to bring up an important, and subtle, issue that I feel should be addressed in this communication and going forward. On pg. 4 line 5, ≪The floe-diameter parameter is a tracer field in CICE, and is transported within each ice category to give the total floe-size distribution at the end of a time step≫. The mean floe diameter, however, does not advect as a tracer.*

$$\vdots$$

*Quite possibly the proper mean floe size advection scheme is unimportant, but as you are the first to introduce this type of model, it is unclear, and is exciting to find out. If future models include a fully-evolving FSD, this fix will no longer be necessary.*

We consider a "representative" diameter in each cell, as opposed to a mean diameter with respect to a distribution. We set the representative diameter to be the mean diameter of a split PDF if breakup occurs, but without attaching the PDF to the cell, as this would be inconsistent with the Steele model, which considers only a single "average" diameter.

Following the wave–ice routine and lateral melt, we transport the representative diameter by:

(i) Setting the floe diameter to be identical for each thickness category, and transporting the floe diameter as an area tracer for the different categories.

(ii) Setting the new representative diameters to be the diameters of the thinnest ice category (cat. 1).

Step (i) is valid with respect to area normalisiation (a delta function in the FSTD, with respect to floe size). Step (ii) is merely a simplifying assumption; however, it does not impact our results, as shown in the figure below. The figure shows a subset of the data from the left-hand panel of manuscript Fig. 2, comparing the mean–monthly ice concentration at the ice edge during January generated by simulations without breakup ($\times$) and with breakup ($\bullet$). Additional results are overlaid for the first 12 simulation years, in which part (ii) uses the diameter of cat. 2 ice ($*$) and cat. 3 ($\square$), neglecting thicker ice categories for clarity and on the basis that thinner ice is most prevalent at the ice edge. Cats. 1–3 give virtually indistinguishable results, indicating that "the proper floe size advection scheme is unimportant" – at least for the metrics we focus on in this investigation.

In the revised manuscript, we have expanded the passage on transport of the representative diameter to include the key points of the above discussion.

[Figure]

Figure: The left-hand panel of manuscript Fig. 2, excluding data from the smaller/larger floe sizes and attenuation rates, and including results in which the representative floe size following advection is set to the floe size of cat. 2 ice ($*$) and cat. 3 ($\square$), rather than cat. 1.

---

## Author Comment (AC3) · 17 Mar 2017

*This paper starts with the CICE sea-ice model and adds waves that break up the sea ice in grid cells near the ice edge, causing that ice to melt more quickly in summer due to greater lateral surface area. In general the paper is well conceived and the mechanism is plausible. My comments (below) are minor.*

We thank the referee for reviewing our paper, and his/her supportive comments and useful suggestions.

*Page 1, Abstract. Total ice volume is reduced by over 500 $km^3$, but this needs to be put into context. What is the volume of the entire ice cover? Page 2, lines 6-7. Same comment as above.*

We now quantify the volume loss in terms of proportion of the total ice volume.

*Page 2, equation (3). What are the units of beta_2 and beta_4? From equation (2), alpha must have units of 1/length. The sea-ice concentration, c, is dimensionless, and omega has units of 1/time, so beta_2 must have units of $time^2$/length, and beta_4 must have units of $time^4$/length. This should be explicitly noted.*

We added units to the values of $\hat{\alpha}_2$ and $\hat{\alpha}_4$ (formerly $\beta_2$ and $\beta_4$; changed to avoid confusion with $\beta_0$ and $\beta_1$ used in the FSD).

*Page 3, line 10. How is a_bk(x) chosen or determined? There is nothing about it in the rest of the paper.*

We now provide a clearer definition of $a_{\mathrm{bk}}$.

*Page 3, lines 13-14. The assumption here is that the floe size distribution (FSD) follows a split power law, with one exponent for floes smaller than a critical size and another exponent for floes larger than the critical size, following Toyota et al (2011). This is a dubious formulation of the FSD. First of all, if one actually looks at figure 9 of Toyota et al (2011), one sees that the FSD is a continuously curving concave-down shape, rather than two power-law regimes. This was noted in earlier work by Herman (2010), who wrote in reference to an earlier paper by Toyota: "However, contrary to how the above authors interpret their results, in both cases the change in slope of the FSD seems rather gradual than abrupt. Instead of a combination of two power laws glued together at a highly arbitrarily chosen floe diameter, another type of distribution would be desirable. It should reflect the observed gradually increasing deviation from a power-law distribution for decreasing floe diameter." Herman, A. (2010), Sea ice floe size distribution in the context of spontaneous scaling emergence in stochastic systems, Physical Review E, 81, DOI: 10.1103/ PhysRevE.81.066123 Furthermore, other researchers have found power-law behavior for the Antarctic FSD in which the exponent changes as the ice edge is approached, but without a critical floe size separating two power-law regimes (Paget et al, 2001; Lu et al, 2008). Returning to the current paper, a much simpler assumption for the FSD would have been a simple power law with one exponent. This would have eliminated the need for four parameters: D_cr, q, gamma_1, and P0. It would be interesting to know whether the results hold up under this simpler (and possibly more realistic) FSD. However, I would not insist that the authors re-do all their calculations, unless it's a simple thing to do (maybe just set D_cr*

*= D_mn and P0 = 0). But they should acknowledge that their results rest on the questionable split power-law formulation of the FSD.*

We added an acknowledgement that other FSD's have been postulated for the transition from small–large floe sizes.

In a preliminary version of the model, we used a fixed breakup diameter $D_{bk} = 30\,\text{m}$ (based on anecdotal observations from our colleagues), and later found that the move to a breakup diameter based on an FSD and the local wavelength produced only small quantitative changes in our results. Therefore, we predict that adjusting the fine details of the in-cell FSD will not significantly impact our findings.

*Page 3, equation (5). In the definition of P0, there is an exponent gamma_1. Is this the same gamma_1 as in equation (4b)? Why should it be the same exponent as in the probability density function? I don't understand the reasoning or the math for the use of gamma_1 here.*

We have corrected $\gamma_0 \to \gamma_1$ and $\gamma_1 \to \gamma_2$ in the paragraph below Eqs. (4a–b).

We set the parameter $\mathbb{P}_0$ so that the proportion $q$ of the floes have diameters greater than the predicted predicted breakup diameter, i.e.

$$1 - q = \int_{D_{mn}}^{D_{pr}} p(d)\ \mathrm{d}d$$

$$= \mathbb{P}_0 + (1 - \mathbb{P}_0)(1 - D_{cr}^{\gamma_1}/D_{pr}^{\gamma_1})$$

$$\Rightarrow \quad \mathbb{P}_0 = 1 - q\left(\frac{D_{pr}}{D_{cr}}\right)^{\gamma_1},$$

as given in the text. We made no changes in response to this part of the comment.

*Page 4, equation (7) and following. What are the units of r_lat and w_lat? What is the value of the time step delta_t?*

We added units for $w_{lat}$ and clarified that $r_{lat}$ is a fraction.

In the first paragraph of § 2, we now explicitly give the time step in terms of the notation $\Delta t$.

*Page 6, lines 8-9, and Figure 2 (left panel). The text and the figure indicate that LESS attenuation of waves results in HIGHER mean ice concentration. I would have thought that less attenuation would allow more wave energy to penetrate into the ice pack and break up the ice, resulting in lower ice concentration. Please explain why less attenuation leads to higher ice concentration, and more attenuation leads to lower ice concentration.*

We corrected the mistake in the text...

*Page 6, line 15. "reducing the attenuation rate increases the impact..." I agree that reducing the attenuation rate SHOULD increase the impact of the waves, but Figure 2 shows that reducing the attenuation rate actually reduces the impact of the waves. The symbols for reduced attenuation rate (upward-pointing triangles) are much closer to the no-break-up case (crosses) than the symbols for increased attenuation rate (downward-pointing triangles). This doesn?t make sense to me.*

. . . and the corresponding mistake in the Fig. 2 caption.

*Page 7, line 8. "volume losses of 0.5 km³ per grid cell" — This doesn't mean anything unless we know how big a grid cell is. On page 2, lines 15-16, we are told that the nominal resolution of the grid is 1 degree in latitude and 1 degree in longitude. The extent of 1 degree of longitude depends on latitude, so the size of a grid cell (in km²) depends on latitude. At the latitude of the Antarctic Circle, 1 degree of longitude is about 44 km. So I calculate that the area of a grid cell is roughly 111 x 44 = 4884 km². A volume of 0.5 km³ of ice spread over such a grid cell is about 10 cm of ice thickness (at 100% concentration) or 20 cm of ice thickness (at 50% concentration). So now I can understand roughly what a volume loss of 0.5 km³ per grid cell means. Please help out the reader by providing this kind of information.*

We now present volume losses per unit area.

*Page 7, lines 20-23. If I understand this correctly, the eastern sector contains mostly first-year ice ("new ice"), with no memory of break-up, while the western sector presumably contains some multiyear ice, which retains memory of break-up?*

We have rewritten this passage to clarify that summer volume losses are carried forward into winter in the western sector only, removing the word 'memory' that may cause confusion.

*Technical Comments*

*Page 1, Abstract, line 2. "Model output shows that WAVE-INDUCED breakup…"*

Added.

*Page 2, equation (1). Cite Thorndike et al (1975), The Thickness Distribution of Sea Ice, JGR.*

Citation added.

*Page 3, line 18. "which is chosen to be equal TO the diameter…"*

Typo corrected.

*Page 3, line 20. Values are given for gamma_1 and gamma_2, but they should probably be gamma_0 and gamma_1. There is no gamma_2 in the equations.*

Corrected: see earlier response.

*Page 3, line 25. propagating through a uniform floe FIELD (?)*

The predicted breakup diameter is calculated as the distance between successive peaks in strain produced by a regular wave propagating along a uniform ice cover of infinite extent.

We added the description "infinitely long" to the text.

*Page 3, equation (6). Inside the integral, "pD" should be "p(D)" i.e. put parentheses around the "D"*

Corrected $pD \longrightarrow p(d)d$.

*Page 4, lines 21-22. In reference to Figure 1, in the left-hand panels showing wave height, it looks to me like the "sharp outer boundaries indicating the latitudes at which data is extracted from the wave model" are at the same latitude in each panel. But the next sentence says, "The boundary is farther north in winter..." I don?t see that the outer boundary of extracted data is farther north in the bottom panel. The outer circular boundary appears to be at exactly the same latitude in both panels. If the authors are referring to an INNER boundary that is several grid cells inside the outer boundary, they should mark it more clearly.*

We added text to clarify that the "sharp out boundaries" we refer to are of non-zero wave heights.

*Page 4, line 32. This sentence should refer to the middle column of Figure 1, just as the previous sentence refers to the left-hand column. Otherwise the reader may not shift her/his attention to the middle column.*

We have amended this sentence.

*Page 5, Figure 1, upper right panel showing concentration change. This panel is a bit too small — it's hard to see the regions of large change. A figure the size of Fig 3 would be better.*

We changed to orientation of the array to maximise the size of the panels.

*Page 7, line 9. "bottom-left panel" should be "bottom-right panel"*

Typo corrected.

*Page 7, line 17. "ice volume per latitude" should be "ice volume per degree of latitude". Similarly in Figure 3 (in the title of the bottom right panel) and in the caption.*

Changes made.

*Page 8, Figure 3. The eastern and western sectors should be marked in the upper panels (the maps).*

Sectors are now marked.

*Page 9, lines 2-3. "If the community judges the impacts..." ? maybe better to say "If further research finds the impacts..."*

We made the suggested change.

*Page 11. "Schwinger" should be "Schweiger"*

Typo corrected.

---

## Author Response (AR1)

[revised manuscript text omitted]

 if  $d \in [D_{\mathrm{cr}}, \infty)$ , where  $\beta_1 = D_{\mathrm{cr}}^{\gamma_1}$ , (5b)

and p(d) = 0 if  $d < D_{mn}$  (Williams et al., 2012). Here,  $D_{mn}$  represents a minimum floe diameter, which is chosen to be equal to the small-floe diameter; for small floes for simplicity;  $D_{cr}$  is a critical diameter marking the transition from small to large floes (found to be in the range 15–40 m by Toyota et al., 2011), and  $\gamma_{\underline{1}0} = 1.15$  and  $\gamma_{\underline{2}1} = 2.5$  are representative exponents for small- and large-floe regimes, respectively (Toyota et al., 2011). The quantity  $\mathbb{P}_0 \in [0, 1]$  weights the distribution towards small floes (large  $\mathbb{P}_0$ ) or large floes (small  $\mathbb{P}_0$ ). Its value is set as

$$\mathbb{P}_0 = 1 - q \left(\frac{D_{\rm pr}}{D_{\rm cr}}\right)^{\gamma_1} \qquad \text{where} \qquad D_{\rm pr} = \lambda/2 \qquad \text{is the predicted breakup diameter,} \tag{6}$$

equal to the distance between successive strain maxima for a regular wave train at the dominant wavelength  $\lambda$  for the spectrum S, propagating through an infinitely long, uniform floe (Williams et al., 2013a; Bennetts et al., 2015), so that a chosen proportion q of floe diameters are greater than  $D_{pr}$ . In the uncommon event that  $D_{pr} < D_{cr}$  then  $\mathbb{P}_0 = 0$ , noting that  $D_{cr}$  approximates the theoretical diameter below which flexural breakup cannot occur (Toyota et al., 2011). The broken-floe diameter  $D_{bk}$  is the mean diameter in a given cell, i.e.

$$D_{\rm bk} = \int_{D_{\rm mn}}^{\infty} p(d) d \, \mathrm{d}d\underline{pD} \, \mathrm{d}D = \frac{\mathbb{P}_0 \gamma_0 \beta_0 (D_{\rm mn}^{1-\gamma_0} - D_{\rm cr}^{1-\gamma_0})}{\gamma_0 - 1} + \frac{(1 - \mathbb{P}_0) \gamma_1 \beta_1 D_{\rm cr}^{1-\gamma_1}}{\gamma_1 - 1}.$$
(7)

The breakup model is applied at the beginning of each CICE time-step, allowing the reduced floe diameters to affect other CICE-model components. The reduced diameters directly affect the contribution of lateral melting fraction of ice that melts laterally,  $r_{\text{lat}}$ , to reducing the ice concentration via the discrete version of Steele (1992)'s model

10
$$r_{\text{lat}} = \frac{\pi \Delta t w_{\text{lat}}}{\mu D},$$
 (8)

which assumes floes in a given cell are identical. Here  $\mu = 0.66$  is a geometric parameter, and  $w_{\text{lat}} = 1.6\Delta T^2 \times 10^{-6}$  (units of distance×time-1) is the rate of lateral melt, in which  $\Delta T$  is the temperature difference of the sea surface above that of the bottom of the ice (set to zero if the difference is negative). The diameter is updated at the end of the thermodynamic routine to account for lateral melt. The floe-diameter parameter is a tracer field in CICE, and is transported within each ice category to

**15 give the total floe-size distribution at the end of a time step.**

During the summer months, when the ice is weaker and towards its minimum extent, waves cause breakup close to the coastline. The existing thermodynamic models in CICE do not increase the diameters of these broken floes fast enough through the winter to create a realistic seasonal cycle for the floe-diameter distribution. Therefore, an ad-hoc floe-bonding scheme is applied, in which the floe diameter in a given cell is doubled if the freezing potential in that cell is positive, up to the maximum

20 diameter  $D_{mx}$ .

5

The representative diameter, D, is transported by: (i) setting the floe diameter to be identical for each of the different thickness categories, and transporting the floe diameter as an area tracer for the different thickness categories; and (ii) setting the new representative diameters to be the diameters of the thinnest ice category (cat. 1). Step (ii) is a non-physical simplifying assumption; tests indicate that this assumption does not affect the concentration changes due to breakup presented in § 3.

**25 3 Results**

30

The model was run from 1979–2010 using input wave data generated by a Wavewatch III model hindcast (Durrant et al., 2013), and atmospheric and oceanic data from the U.S. National Center for Environmental Prediction's Climate Forecast System Reanalysis (NCEP's CFSR, Saha et al., 2010). The minimum and critical floe diameters are set as  $D_{mn} = 5 \text{ m}$  and  $D_{cr} = 30 \text{ m}$ , and, following breakup, the proportion q = 0.05 of floe diameters are set to be greater than the predicted breakup diameter  $D_{pr}$ .